# Glycobiology of Cancer: Sugar Drives the Show

**DOI:** 10.3390/medicines9060034

**Published:** 2022-05-24

**Authors:** Jhenifer Santos dos Reis, Marcos André Rodrigues da Costa Santos, Daniella Pereira Mendonça, Stefani Ingrid Martins do Nascimento, Pedro Marçal Barcelos, Rafaela Gomes Correia de Lima, Kelli Monteiro da Costa, Celio Geraldo Freire-de-Lima, Alexandre Morrot, Jose Osvaldo Previato, Lucia Mendonça Previato, Leonardo Marques da Fonseca, Leonardo Freire-de-Lima

**Affiliations:** 1Instituto de Biofisica Carlos Chagas Filho, Universidade Federal do Rio de Janeiro, Rio de Janeiro 21941-170, Brazil; jheniferreis@biof.ufrj.br (J.S.d.R.); rodrigues8mr@gmail.com (M.A.R.d.C.S.); daniella_pm14@hotmail.com (D.P.M.); stefaniingrid77@gmail.com (S.I.M.d.N.); pedrombarcelos73@gmail.com (P.M.B.); rafaelagclima102@gmail.com (R.G.C.d.L.); kellimc85@gmail.com (K.M.d.C.); celio@biof.ufrj.br (C.G.F.-d.-L.); previato@biof.ufrj.br (J.O.P.); luciamp@biof.ufrj.br (L.M.P.); lfonseca@biof.ufrj.br (L.M.d.F.); 2Instituto Oswaldo Cruz, FIOCRUZ, Rio de Janeiro 21040-360, Brazil; alexandre.morrot@ioc.fiocruz.br; 3Faculdade de Medicina, Universidade Federal do Rio de Janeiro, Rio de Janeiro 21044-020, Brazil

**Keywords:** glycosylation, cancer, glycosyltransferases, glycome, glycomedicine

## Abstract

Cancer development and progression is associated with aberrant changes in cellular glycosylation. Cells expressing altered glycan-structures are recognized by cells of the immune system, favoring the induction of inhibitory immune processes which subsequently promote tumor growth and spreading. Here, we discuss about the importance of glycobiology in modern medicine, taking into account the impact of altered glycan structures expressed in cancer cells as potential glycobiomarkers of disease, as well as on cancer development and progression.

The hottest topic in the field of molecular biology during the 1970s and 80s was understanding the flow of information solely between DNA, RNA and proteins [1]. The central dogma of molecular biology states that once that information takes the form of proteins, it cannot be taken back to nucleic acid [2]. Although disproved by the discovery of prion-mediated heredity, its reflections can still be perceived to this day [3]. Clear examples are the illustrations of cell membranes in classical textbooks of biochemistry and cell and molecular biology, where little is discussed about the importance of glycoconjugates. Thanks to scientific and technological advances, nowadays it is well accepted that cell-surface and/or secreted glycomes reflect overall cellular status in health and disease [4]. The term glycome refers to the complete repertoire of glycomolecules decorated with carbohydrate chains, or glycans, that are covalently linked to lipid or proteins [5]. Glycosylation is a highly dynamic and finely regulated process involving a complex biological apparatus whose components are spread into different cellular compartments, such as nucleus, cytoplasm, endoplasmic reticulum, Golgi and lysosomes [6,7,8]. It is estimated that 3–4% of the human genome encodes elements of the glycoconjugate biosynthesis machinery. Among such components we can find enzymes, which are generically called glycosyltransferases and glycosidases, chaperones, sugar transporters and donors, as well as other molecules necessary for the modification of proteins or lipids with carbohydrates [8].

Glycoconjugates are found on the cell surface of all living organisms, and play essential roles in mediating protein-receptor signaling, cell–cell and cell–matrix interactions, and appropriate protein folding and maturation during translation [9]. In fact, changes in glycosylation can modulate inflammatory responses, enable viral immune escape, promote cancer cell metastasis or regulate apoptosis [10]. New insights into the structure and function of the glycome can now be applied to therapy development and could improve our ability to fine-tune immunological responses and inflammation, optimize the performance of therapeutic antibodies and boost immune responses to chronic diseases, such as cancer [11,12]. These examples illustrate the potential of the emerging field of glycomedicine, which communally aim to clarify the function of glycans in person-to-person and between-population discrepancies in disease vulnerability and response to health interventions such as vaccines, nutrition and drugs [13].

Targeting immune checkpoints to improve the outcome of cancer patients is an ongoing discussion in oncobiology. However, few patients have shown long-term benefits from currently used CTLA-4 and PD-1/PD-L1 inhibitors [14,15,16]. Therefore, new strategies are needed to increase the long-term remission after cancer immunotherapy. Over the past ten years, numerous studies have shown that glyco-immune checkpoints can be used as new targets for cancer immunotherapy. They are well-defined as immunomodulatory pathways, including interactions between glycan-binding proteins or lectins with glycan epitopes [14,17,18]. The most prominent pathways involve the immune and vascular programs triggered by galectins [19,20,21,22], as well as the sialo-glycan-Siglec axis [14,23,24,25]. In both cases, inhibitors are already being successfully tested in clinical trials [19,26]. This confirms that advances in the field of glyco-immunology will permit us to improve cancer immunotherapy and help many patients.

Regarding the effects of glycoconjugates in cancer cells, it has been well documented that malignant transformation and tumor progression correlate with aberrant changes in cellular glycosylation [6,12,27]. In cancer cells, *O*-linked glycans are characterized to present immature and/or truncated structures due to reduced expression and/or activity of specific glycosyltransferases, such as beta 1,3-glalactosyltransferase [28] and core 2 beta-1,6-N-acetylglucosaminyltransferase (C2GNT), contributing to the accumulation of altered glycan structures such as Tn and sialyl-Tn antigens [29] and T-antigen and T-sialyl antigen [30]. In contrast, *N*-linked glycans in cancer cells are characterized by being long, branched, and hypersialylated [30]. When it comes to *N*-linked glycans, however, there is more to the story. Many groups show an abundance of long, branched, and hypersialylated structures [30,31,32,33,34,35], while others report high mannose structures [36,37] or even both [38]. One particular study points to high mannose structures being prevalent in the primary tumor, while branched sialylated epitopes are found in metastatic foci [39]. These findings may suggest that just like it’s very hard to find two cancer patients suffering from the exact same disease, glycosylation patterns may vary depending on the precise mutations occurring simultaneously on the cancer cell.

For a long time, such structures were used only for diagnostic purposes. However, many research groups have since demonstrated that structurally altered glycoproteins are able to modulate various events linked to the progression of different types of cancer [40]. Recent studies have demonstrated that altered glycosylation of proteins that make up the glycocalyx may be recognized by immune cells, leading to induction of inhibitory immune processes, which subsequently drive tumor growth and metastasis [41]. Several studies developed by our research group demonstrated that *O*- and *N*-linked unusual glycan structures govern phenomena associated to the epithelial–mesenchymal transition (EMT) process, as well as the acquisition/maintenance of the multidrug resistance (MDR) phenotype [12,34,35,42,43,44,45,46,47,48,49,50]. MDR phenotype and the acquisition of metastatic properties by cancer cells are known as the main obstacles to the treatment of different types of cancer [51]. Over twenty-five years ago, these events were studied as independent phenomena. However, it is now well established that both are necessary to be investigated together [35], since numerous papers have demonstrated that glycan structures present strong impact on the MDR phenotype [12,52,53]. Although several studies have already described that the emergence of aberrant glycan structures is strictly related with the activation of both molecular pathways linked to EMT process and the emergence of MDR phenotype, little is known about how such glycan structures might connect these two multifactorial events. The developing of glycosyltransferase knockouts mice has confirmed that pathological phenotypes may be triggered in vivo by genetic manipulation of glycans [54], demonstrating that proteins decorated with aberrant glycan structures are promising drug targets for treating various diseases, including cancer.

In recent works, we have demonstrated for the first time that alterations in glycosylation in tumor cells chronically exposed to chemotherapeutic agents, are able to connect both MDR phenotype and EMT process, since in addition to presenting changes in the expression and/or activity of efflux pumps belonging to the ABC superfamily (ABCB1, ABCC1 and ABCG2) [55,56], the chemoresistant human cancer cell lines also showed increased cell motility, as well as altered expression of epithelial–mesenchymal markers, when compared with their normal counterparts [34,35]. These findings confirm the idea that both accretion of MDR phenotype and the activation of EMT process, which have been considered indispensable for invasion and metastasis [57,58,59], are deeply linked with unusual glycan structures expressed by transformed cells. In our previous study we also observed that the chronic exposure to non-lethal concentrations of cisplatin induced the expression of an isoform of fibronectin (FN), so called oncofetal FN (onf-FN) [35], which may be found in transformed cells, and embryonic samples, but is absent in normal tissues [6]. onf-FN was also described by Hakomori’s group in cancer cells undergoing EMT [42,43], but its role in many events linked to cancer progression, including the acquisition of drug-resistant phenotype, is still unknown. Taken together, it has become clear that further investigation in this area may offer new diagnostic biomarkers and therapeutic targets to combat this devastating disease, which while no longer a death sentence, is still considered potentially fatal if not diagnosed early.

## Data Availability

Not applicable.

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
