# Peer review of "Glycobiology of Cancer: Sugar Drives the Show"

_medicines, 2022, doi:10.3390/medicines9060034_

Round 1
Reviewer 1 Report
Glycobiology is a fast-growing and hot topic and cancer researches. This opinion provides overview of current knowledge and many recent works. Publication is suggested after the following editing.
Major:
1) The phrase "atypical glycans" appeared five times in the paper, including once in the abstract. To my knowledge, this is no clear definition (or even hotly debated) of "atypical glycans" in the field. Synthesis of glycan is not template-driven, therefore defining "atypical" is tricky and often sample-based. I would suggest the author the change the term, or make conservative definition of the phrase in the paper;
2) The second paragraph is too long, need to be splitted into 3~4 short ones based on the different topic;
Minor:
1) The recent advance glyco-immune checkpoints for cancer immunotherapy should be discussed (line 49~53);
2) The statement of "In contrast, N-linked glycans in cancer cells are characterized by being long, branched, and hypersialylated" (line 62-63) is arguably limited. Many cancer cells demonstrate de-sialyation and dominated by high-mannose. The community is in the process of gathering systematic data. I would suggest an tune-down tone in the discussion of cancer glycomic data. For example, "N-linked glycan in XXX cancer cells are changed toward XXX".
Author Response
Thank you very much for your time and comments. Your suggestions will certainly contribute to the improvement of the manuscript.
- Although the term atypical glycans is udes in several papers (please, see the references PMID 27007155, PMID 33704376, PMID 27500424, PMID 29502191, PMID 33135073 and PMID 33539510), we have modified the expression in the text, (now lines 21, 71, 95, 110 and 112).
- As sugested, the second paragraph has been split in short ones (now starting in lines 43, 55 and 66).
- A new sentence describing recent advance in glyco-immune checkpoints for cancer immunotherapy has been added to the text (now lines 55-65).
- As suggested by the reviewer, we improved the discussion on N-glycans in tumor cells, and added new references. (now lines 74-80).
Reviewer 2 Report
I recommend ensuring latest publications pertaining to the interplay between MDR and EMT and glycosylation are included.
Check spelling of essential in line 44
Please explain this cisplatin study in a more detailed manner... "chemoresistant human cancer cell lines present increased cell motility, as well as altered expression of epithelial-mesenchymal markers when compared with its normal counterparts"...being more specific will emphasize the topic of the paper more
I think the background information should be decreased and more emphasis should be laid on the precise relationship between glycosylation, EMT, and MDR
The article could be made easier to read
Overall, great topic considering the extreme relevance of postranslational modifications in cancer
Author Response
Thank you very much for your time and comments. Your suggestions will certainly contribute to the improvement of the manuscript.
- As suggested by the reviewer, new references regarding EMT, MDR penotype and glycosylation were added to the manuscript (please, see references 44, 45, 46, 47, 48, 49 and 50).
- The spelling has been checked (now line 43).
- As suggested by the reviewer, the sentence "chemoresistant human cancer cell lines present increased cell motility, as well as altered expression of epithelial-mesenchymal markers when compared with its normal counterparts" has been discussed in more details (now lines 102-111).
- The manuscript has been restructured, and as suggested by the reviewer, we emphasize the relationship between MDR phenotype, EMT process and glycosylation.